# Lactate in Heart Failure

**DOI:** 10.3390/ijms26146810

**Published:** 2025-07-16

**Authors:** Piotr Gajewski, Michał Maksymilian Wilk, Krzysztof Aleksandrowicz, Beata Ponikowska, Robert Zymliński

**Affiliations:** 1Institute of Heart Diseases, Wroclaw Medical University, 50-556 Wrocław, Poland; krzysztof.aleksandrowicz@umw.edu.pl (K.A.); robertzymlinski@gmail.com (R.Z.); 2Student Scientific Organization, Institute of Heart Diseases, Wroclaw Medical University, 50-556 Wrocław, Poland; michalwilk200218@wp.pl; 3University Center for Physiotherapy and Rehabilitation, Faculty of Physiotherapy, Wroclaw Medical University, 50-556 Wrocław, Poland; 4Department of Pathophysiology, Wroclaw Medical University, 50-556 Wrocław, Poland; beata.ponikowska@umw.edu.pl

**Keywords:** heart failure, lactate, acute heart failure, chronic heart failure

## Abstract

This narrative review explores the multifaceted role of lactate in heart failure (HF), focusing on its diagnostic and prognostic significance in both acute and chronic HF. Lactate functions not only as a marker of hypoperfusion and anaerobic metabolism but also as an active metabolic substrate. In acute HF, elevated lactate levels often signal circulatory insufficiency and predict adverse outcomes. In chronic HF, especially HFpEF, lactate dynamics during exercise reflect metabolic inefficiency and correlate with functional impairment. This review emphasizes the dual nature of lactate and discusses its potential utility in risk stratification and therapeutic guidance.

## 1. Introduction

Heart failure (HF) is a chronic and progressive clinical syndrome characterized by structural and/or functional cardiac abnormalities. It can manifest as either HF with reduced ejection fraction (HFrEF) or HF with preserved ejection fraction (HFpEF), both conditions leading to increased cardiac filling pressures at rest and during exertion. Symptoms of HF include dyspnea and fatigue, often accompanied by signs such as pulmonary rales and peripheral edema [1].

The increasing prevalence of HF, now affecting approximately 2% of the adult population in developed countries, poses a significant public health concern. This rise is driven by improved survival rates for various cardiac conditions, demographic shifts, and better long-term outcomes associated with HFrEF treatment. Acute heart failure (AHF) is a prevalent cause of unplanned hospital admissions among individuals over 65, complicating management due to its heterogeneous nature, which often masks the underlying pathophysiology [2,3,4,5,6,7].

A critical aspect of AHF is hypoperfusion, defined as inadequate blood flow to meet the metabolic needs of tissues. In HF, hypoperfusion can occur due to reduced cardiac output, often exacerbated by increased filling pressures that lead to systemic congestion and elevated vascular resistance. As a result, organs and tissues may experience ischemia, leading to further dysfunction across multiple systems, including the kidneys, liver, and gastrointestinal tract. This multi-organ dysfunction significantly impacts patient outcomes and increases the risk of adverse events [8,9,10].

Hypoperfusion not only presents as cardiogenic shock but can also manifest in subclinical ways, affecting the quality of life and functional capacity of chronic heart failure (CHF) patients. Symptoms such as fatigue, confusion, and reduced exercise tolerance may be attributed to hypoperfusion, resulting from transient drops in perfusion pressure. Furthermore, the dysregulation of neurohormonal systems in HF, including increased levels of catecholamines and renin–angiotensin–aldosterone, can further exacerbate hypoperfusion by increasing vascular resistance and impairing blood flow distribution [11,12,13,14].

The 30-day readmission rate for AHF can be as high as 24%, escalating to 50% within six months. Patients who experience rehospitalization due to circulatory system diseases within the first 90 days after discharge face an increased risk of mortality. These statistics highlight the urgent need for improved management strategies for acute decompensated heart failure (ADHF) and underscore the critical role of preventing persistent fluid overload post-discharge, which can elevate the risk of readmission, morbidity, and mortality [6].

Given the evolving understanding of HF pathophysiology, this narrative review aims to synthesize current knowledge on the role of lactate in both acute and chronic HF, emphasizing its diagnostic, prognostic, and potential therapeutic implications.

Lactate is a biomarker that has garnered increasing attention for its potential role in the diagnosis and prognosis of HF. In this narrative review, we explore the physiological and pathophysiological significance of lactate in both acute and chronic HF. Over the past decades, numerous studies have investigated the role of lactate, particularly in critically ill patients, shedding light on its complex functions in metabolic regulation, tissue perfusion, and disease progression. This article aims to synthesize current knowledge and highlight the clinical relevance of lactate in the context of HF. However, hyperlactatemia, defined as blood lactate >2 mmol/L, can arise from multiple causes, such as reduced oxygen delivery (tissue hypoxia), impaired oxygen extraction, peripheral shunting, stress, and increased adrenergic activity. In acutely ill patients, it is challenging to determine the relative contribution of each factor [15,16,17]. Lactate is primarily a product of glycolysis and plays a pivotal role in the regulation of various tissues and organs, particularly within the cardiovascular system. Traditionally, lactate is considered a by-product of anaerobic cell metabolism, signaling tissue hypoxia, most often due to impaired peripheral perfusion. However, the interpretation of systemic lactate levels is much more complex and reflects a fine-tuned balance between lactate production and its elimination. Under normal conditions, the former occurs in most tissues, mainly in skeletal muscle, while lactate is rapidly eliminated, mainly in the liver and kidneys [18,19,20] as shown on Figure 1. 

Lactate, as one of the well-known metabolic compounds, plays a crucial and multifaceted role in the body’s physiological response to increased demand for adenosine triphosphate (ATP) and insufficient oxygen supply, particularly in critical and acute conditions such as HF and sepsis. Its production, primarily stemming from glycolytic pathways, provides an essential mechanism for maintaining energy homeostasis during periods of hypoxia or ischemia. This process allows cells to continue generating ATP when oxygen levels are inadequate, highlighting lactate’s crucial function in energy metabolism.

Moreover, contemporary perspectives suggest that lactate should not be merely viewed as a waste product of anaerobic metabolism; instead, it should be recognized as a significant indicator of the body’s metabolic state and a key player in various physiological processes. For instance, lactate participates actively in the Cori cycle, a biochemical pathway that facilitates the recycling of lactate produced in muscles back into glucose in the liver, thereby sustaining energy production across different tissues (Figure 2).

In cardiac physiology, lactate serves a vital role as both a substrate and a metabolic signal [21,22]. The heart can utilize lactate efficiently, drawing on it as an alternative energy source, especially when the supply of fatty acids is compromised during episodes of disease exacerbation. This metabolic flexibility is crucial for maintaining cardiac function under stress [23,24].

Furthermore, recent research emphasizes the potential of lactate monitoring as a valuable tool in clinical practice, aiding in medical decision-making and risk stratification for patients with heart failure and other cardiovascular diseases. By understanding the dynamic role of lactate in metabolism and its implications for patient management, healthcare professionals can better tailor therapeutic approaches to mitigate complications and improve outcomes. Thus, recognizing lactate’s diagnostic and prognostic value can significantly enhance strategies for managing coronary artery disease and HF, ultimately leading to improved patient care and survival rates.

This expanded version provides more context and depth regarding lactate’s role and significance in metabolism, especially in the context of HF.

## 2. Acute Heart Failure (AHF)

Acute heart failure (AHF) may be associated with low-tissue perfusion and/or hypoxaemia leading to increased lactate levels and acid–base perturbations [25]. Lactate is a marker of illness severity and is independently associated with in-hospital mortality in AHF. Few data are available on the clinical significance of elevated lactate levels [9,20,26,27].

In the context of AHF, the prevailing hypothesis posits that increasing lactate concentrations signify a shift towards anaerobic glycolysis at the cellular level, serving as an indicator of end-organ hypoperfusion and/or hypoxia [28,29].

Crucially, however, hyperlactatemia can arise independently of tissue oxygen availability due to an accelerated rate of aerobic glycolysis. The lactate anion is continuously produced and utilized in a variety of cells under aerobic conditions, playing a significant role as an energy substrate during periods of heightened metabolic demand. The stimulation of aerobic glycolysis, coupled with muscle protein catabolism—resulting in the release of amino acids—leads to increased pyruvate production, which subsequently converts into lactate. Under these circumstances, the rate of lactate production may surpass the oxidative capacity of the mitochondria [30,31].

The concentration of lactate is contingent upon the equilibrium between the glycolytic rate and the subsequent mitochondrial metabolism. An increased mitochondrial volume density and preserved mitochondrial function can sustain lower lactate levels even under elevated metabolic demands; conversely, mitochondrial intrinsic dysfunction, as observed in HF, can contribute to lactate accumulation. Therefore, hyperlactatemia frequently reflects an upsurge in metabolic rate and sympathetic nervous system activation [32,33].

Catecholamines, particularly epinephrine—more so than norepinephrine—contribute to an elevation in plasma lactate concentration by potentiating the metabolic mechanisms. Thus, neurohormonal activation, especially heightened adrenergic stimulation, can exacerbate lactate production [15,17,34]. The physiological effect of lactate surge during exposure to neurohormonal stimulation via catecholamines is caused by the increase in productivity of glycolysis and glycogenolysis.in skeletal muscles. Simultaneously, increased oxygen consumption is observed, which creates a preferable position for anaerobic metabolism, further enhancing the glycolysis share in maintaining the energetic balance in myocytes [15,34]. Anaerobic glycolysis consequently leads to an upsurge in blood lactate concentration; however, the state of hyperlactemia can be buffered thanks to the Cori cycle, where excessive lactate is utilized as metabolic fuel for hepatic gluconeogenesis [34]. Interestingly, spare lactate might paradoxically play an essential role in regulating cardiomyocyte metabolic patterns involved in providing energetic stability of the whole heart, what is associated with the modulation of NAD^+^/NADH^+^ H^+^ status although the concentration range for this beneficial effect seems to be insignificant and adverse effects of accumulating lactate overtake potential gains rapidly.

An understanding of the causes of lactate accumulation is conducive to personalized treatment strategies.

Multiple studies have shown that elevated lactate levels—particularly ≥2 mmol/L—are strongly associated with increased short- and long-term mortality in AHF. Persistent hyperlactatemia further portends poor outcomes, emphasizing the utility of serial lactate assessment for risk stratification [20,35].

Another investigation explored the effects of persistent hyperlactatemia in HF patients, revealing that those with sustained high lactate levels at both admission and 24 h later had an increased risk of mortality within a year [36]. Additionally, the combination of elevated lactate levels and intracellular iron deficiency significantly worsened survival outcomes compared to patients without this condition [37].

Another study confirms that arterial lactate levels are a significant independent predictor of all-cause hospital mortality in patients with AHF in the ICU. Higher lactate levels were directly associated with an increased likelihood of hospital mortality, indicating that as lactate levels rise, the risk of death also increases. Additionally, the predictive value of lactate levels can be enhanced by combining them with the Simplified Acute Physiology Scale (SAPS) II. The analysis revealed that the association between lactate levels and hospital mortality was mediated by the presence of respiratory failure, highlighting the importance of individualized assessments for patients experiencing this complication. Furthermore, NT-proBNP was identified as a mediator in the relationship between lactate levels and hospital mortality, suggesting its potential role in the underlying mechanisms of AHF [38].

Although increased lactate levels generally indicate a poor prognosis, lactate also serves as a crucial energy source for the heart, particularly during stress. Research has shown that failing hearts utilize more lactate and ketones for energy. Clinical trials have demonstrated that sodium lactate infusions can improve cardiac output and function without harming organ systems [39]. Furthermore, animal studies suggest that lactate uptake may enhance myocardial energy metabolism in congestive HF [40].

While elevated lactate levels typically signal poor outcomes in heart failure, the dual role of lactate—as both a marker of risk and a metabolic fuel—highlights the need for further research to clarify whether lactate is ultimately harmful or beneficial in this context.

## 3. Chronic Heart Failure (CHF)

Data on lactate in CHF are limited and primarily based on studies from many years ago. The reviewed studies indicate that patients with HF accumulate blood lactate at lower absolute workloads compared to healthy individuals, and there is evidence suggesting that this reliance on anaerobic metabolism at low workloads appears to be exaggerated with increasing disease severity [41]. However, a closer examination of the data reveals that the anaerobic threshold (AT), that can be described as exercise intensity at which lactate begins to accumulate in the blood, indicating a shift toward anaerobic metabolism, occur in patients with HF occurs at relatively high percentages of total working capacity or VO2 max. AT is reported to occur at about 60–70% of VO2 max in patients with mild to severe HF, with similar percentages (65–72%) observed, showing a trend toward AT occurring at higher percentages of peak VO2 with decreasing functional capacity. Additionally, data show that in younger, healthy individuals, AT occurs at slightly lower levels than in patients with HF (64–68% of peak VO2) [41,42,43,44,45,46,47,48].

There is considerable evidence that exercise training delays blood lactate accumulation during submaximal exercise, which seems to be due to the increased oxidative capacity of skeletal muscle. Increases in mitochondrial size and number and oxidative enzyme activity appear to be primary contributors to these improvements, while evidence for increased capillarization and changes in fiber type from glycolytic type IIb to oxidative type I is less convincing [46,47].

Most reviewed studies reported that lactate levels in patients with HF were significantly lower than those in healthy individuals at maximal or peak exercise capacity, with the only exception observed during exercise using smaller muscle mass. Peak lactate values have also been reported to decrease with decreasing functional capacity in patients with HF. Only one study reported increases in peak lactate values after exercise training, and this result seems to be due to the different exercise modes (combined endurance/resistance) used in that study [49].

The study, conducted by Luigi Adamo et al., focuses on the prevalence of lactic acidaemia in patients with advanced HF and low cardiac output. It reveals a critical insight into the metabolic alterations observed in these patients, specifically the unexpectedly low prevalence of lactic acidaemia, which stands at about 25%. This cohort, characterized by significantly depressed cardiac output and reduced oxygen delivery to peripheral tissues, challenges pre-existing notions that such cardiac impairment consistently results in a metabolic shift from aerobic to anaerobic processes. Notably, out of 89 individuals studied, only 24 exhibited elevated plasma lactate levels, and importantly, there was no observed correlation between cardiac index and peripheral lactate concentrations. This finding suggests that increased lactate production occurs predominantly as a late-stage phenomenon in advanced HF scenarios [50]. Lactate production and clearance dynamics underscore a complex metabolic interplay where lactate is typically produced by most body tissues and efficiently cleared by the liver, which manages about 70% of lactate metabolism, supplemented marginally by renal clearance. In HF, lactic acidosis, defined as hyperlactatemia with blood pH < 7.35, is conventionally divided into type A, associated with impaired tissue perfusion, and type B, stemming from reduced clearance. This study’s retrospective analysis points towards a dual presence of both acidosis types among HF patients. The elevated lactate levels observed are not solely due to enhanced production by hypoxic peripheral tissues but may also reflect impaired hepatic function, as indicated by higher INR and AST readings and lower albumin levels in patients [18,21,22,23,50]. While the investigation does not conclusively identify all determinants of lactic acidemia, it illuminates the multifaceted nature of metabolic dysregulation in HF. It underscores the importance of dissecting these metabolic nuances to inform the development of novel therapeutic strategies. The research acknowledges certain limitations, including its focus on patients with more severe conditions, which might not fully represent the metabolic state across all metabolizing tissues. Furthermore, systemic lactate measurements may overlook localized increases in lactate production. Nonetheless, the study advocates for a reevaluation of traditional models managing HF, highlighting the urgency for ongoing research to refine and validate these models in the context of modern patient demographics and clinical presentations [18].

The latest data show that patients with heart failure with preserved ejection fraction (HFpEF) exhibit elevated resting lactate levels and a significant increase in indexed lactate during exercise compared to healthy volunteers. This lactataemia in HFpEF patients is driven by key physiological changes, including chronotropic incompetence and a lower cardiac index resulting from reduced stroke volume and heart rate during physical activity. Furthermore, individuals with HFpEF demonstrate altered peripheral oxygen dynamics, characterized by higher mixed venous oxygen saturation, reduced oxygen delivery, and a decreased oxygen extraction ratio, indicating an earlier transition to anaerobic metabolism in response to compromised cardiac output.

Research conducted by Nan Tie et al. indicates that HFpEF affects not only cardiac energetics but also peripheral factors, such as altered mitochondrial content and reduced skeletal muscle oxidative capacity. These changes correlate with decreased exercise capacity, shifting the perspective on lactate from being merely an indicator of anaerobic metabolism to a substrate and signaling molecule [51].

Persistent lactate accumulation may arise from an imbalance between lactate production and clearance, influenced by skeletal muscle composition and metabolism, as well as conditions like congestive hepatopathy that further hinder lactate clearance. The study reveals that patients with HFpEF experience lactataemia at relatively low peak workloads, suggesting they may encounter significant lactate elevations during daily activities. This understanding emphasizes the complex interplay of cardiac and peripheral factors in HFpEF and its impact on exercise limitations in these patients [51]. Emerging evidence suggests that sodium-glucose co-transporter 2 (SGLT2) inhibitors may favorably modulate lactate metabolism. These agents enhance mitochondrial function and reduce oxidative stress, potentially improving lactate clearance and utilization in HF patients [52]. The most important findings regarding the role of lactate in the context of HF are included in Table 1.

## 4. Conclusions

Lactate is emerging as both a biomarker of metabolic stress and a therapeutic consideration in HF. In AHF, elevated lactate levels correlate with hypoperfusion and worse prognosis. In chronic HF, particularly HFpEF, lactate reflects impaired metabolic adaptation to exertion. Its dual role—as a risk marker and an energy substrate—warrants further exploration. Integrating lactate into clinical practice may enhance risk stratification and help tailor patient-centered interventions.

The distinctions between lactate’s roles in AHF and CHF are significant. In AHF, rising lactate concentrations are closely associated with worsening clinical status and increased mortality risk. The shift toward anaerobic metabolism and the resultant lactate accumulation reflect underlying circulatory insufficiencies that necessitate immediate and targeted therapeutic interventions. Understanding these dynamics is critical for healthcare providers, as it aids in risk stratification and the management of patients experiencing acute decompensation.

Conversely, in the realm of CHF, the accumulation of lactate at lower exercise workloads illustrates impaired metabolic efficiency and can be indicative of the disease’s progression. The correlation between lactate levels and exercise capacity, particularly in patients with preserved ejection fraction (HFpEF), highlights the need for a more nuanced approach to patient evaluation and management. This understanding emphasizes the importance of considering peripheral factors and systemic responses when addressing the metabolic derangements associated with HF.

Furthermore, as research continues to elucidate the multifaceted role of lactate in HF, there is an emerging recognition of its potential as both a biomarker and a therapeutic target. The dual nature of lactate—as a signal of metabolic distress and as a substrate for energy production—poses intriguing questions about how best to leverage this knowledge in clinical practice. Future studies should focus on elucidating the mechanisms underlying lactate metabolism in HF, exploring the implications of lactate clearance in organ function, and assessing the benefits of therapeutic strategies involving lactate administration.

In conclusion, integrating lactate measurements into routine clinical assessments has the potential to enhance management strategies for patients with HF, improve individual outcomes, and contribute to the development of personalized therapeutic approaches. By recognizing and addressing the complex role of lactate within the broader context of heart disease, clinicians can better navigate the challenges of HF management, ultimately leading to improved care and survival rates for patients.

## Figures and Tables

**Figure 1 ijms-26-06810-f001:**
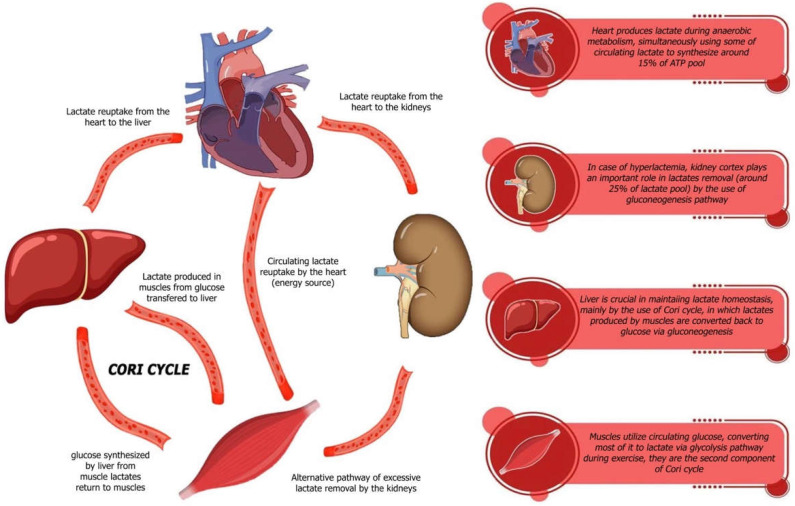
Role of lactate in energy metabolism under hypoxic conditions.

**Figure 2 ijms-26-06810-f002:**
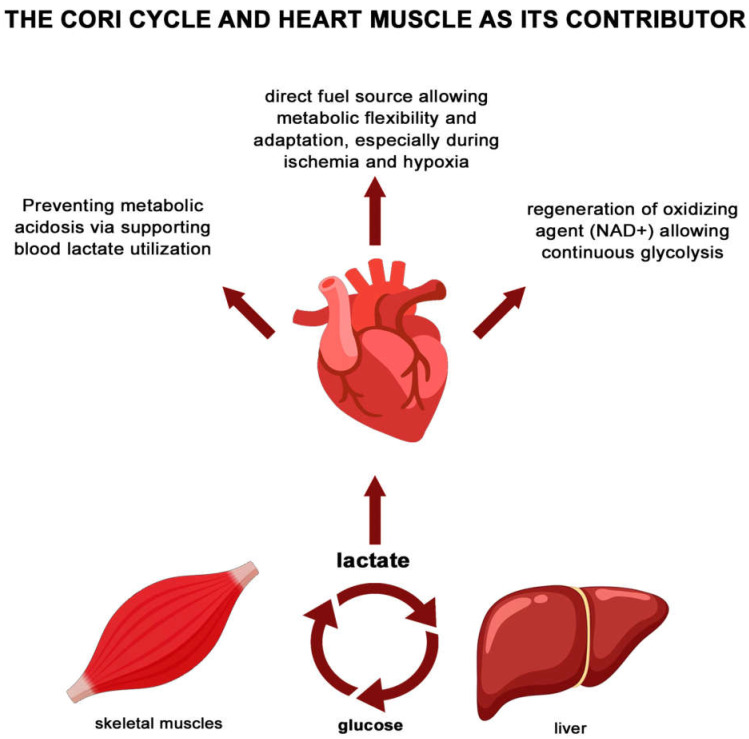
Lactate as an energy substrate and signaling molecule in cardiac physiology.

**Table 1 ijms-26-06810-t001:** Summary of Key Clinical Studies Investigating Lactate as a Prognostic and Metabolic Marker in Acute and Chronic Heart Failure.

Study (Author Year)	Hf Type	Sample Size/Population	Lactate Threshold	Key Clinical Outcomes/Findings
Weber et al., 1985 [41]	CHF	Patients with mild to severe CHF undergoing cardiopulmonary exercise testing; n = not stated	Anaerobic threshold at 60–70% of VO2 max	HF patients show earlier shift to anaerobic metabolism; lactate accumulates at lower workloads compared to healthy individuals
Nalos et al., 2014 [39]	AHF	Pilot RCT, n = 40 patients with acute heart failure receiving sodium lactate vs. control	Exogenous sodium lactate infusion	Sodium lactate infusion improved cardiac output and hemodynamics without negative effects on organ function
Kawase et al., 2015 [20]	AHF	ICU patients with acute decompensated HF; n = 113	>3.2 mmol/L	High lactate at admission significantly predicted early mortality, independent of other clinical variables
Adamo et al., 2017 [18]	CHF (Advanced)	n = 89 patients with advanced HF and low cardiac output	~25% had elevated lactate	Despite low cardiac output, only 25% had hyperlactatemia; suggests lactate elevation is a late metabolic event in CHF
Zymliński et al., 2018 [9]	AHF	n = 312 AHF patients, mostly normotensive, no overt shock	≥2.0 mmol/L	Elevated lactate associated with higher 1-year mortality and evidence of cardiac/hepatic injury; useful for early risk stratification
Gjesdal et al., 2018 [35]	AHF	n = 188 patients with MI complicated by HF but without cardiogenic shock	>2.5 mmol/L	Increased lactate strongly predicted 30-day mortality; highlighted lactate’s role in post-MI prognosis
Biegus et al., 2019 [36]	AHF	n = 259 AHF patients admitted to ICU	Persistent elevation > 24 h	Patients with persistently elevated lactate had worse in-hospital and 1-year outcomes than those whose lactate normalized
Biegus et al., 2019 [37]	AHF	n = 405 AHF patients with assessment of iron status	Any elevation	Combined elevated lactate and intracellular iron deficiency dramatically worsened prognosis compared to patients without both risk factors
Hu et al., 2022 [38]	AHF	n = 1201 ICU patients with AHF	Continuous (range not specified)	Lactate was an independent predictor of in-hospital all-cause mortality; predictive value increased when combined with SAPS II and NT-proBNP
Nan Tie et al., 2024 [51]	HFpEF (CHF)	n = 36 HFpEF patients vs. 19 healthy controls in exercise testing	Elevated resting and exercise lactate	HFpEF patients showed early lactate rise during low workloads due to chronotropic incompetence and impaired oxygen extraction; reflects early anaerobic switch during daily activity

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
