# Peer review of "Lactate in Heart Failure"

_ijms, 2025, doi:10.3390/ijms26146810_

Round 1

Reviewer 1 Report

Comments and Suggestions for Authors

Authors of ms ijms-3617044 review the literature on lactate in heart failure. In fact, reading the whole review it turns out that lactate has been studied mostly in acute heart failure, since it is eminently a marker widely used in critically ill patients. Few data are available on chronic HF, and mostly on lactate concentrations in response to exercise. That’s why the title may be more informative by adding “acute”. The data on chronic HF are an add-on.

The figures are clear and informative.

Having presented evidence on the clinical utility of lactate in refining prognosis of patients with acute HF, the Authors state that knowing plasma lactate may help in designing treatment strategies of patients with HF, but data are not presented to support this statement.

Overall, the present review does not contribute to the advancement of knowledge in the use of lactate in HF, when compared to a previous review (ref#61) published in 2017. The reader may benefit of an assessment on what do we know, when compared to ref#61? What has research on lactate in HF produced in these last 8 years?

Author Response

Thank you very much for your comment and thorough analysis of our manuscript. I agree that the role of lactate has been very well described in advanced heart failure as a prognostic marker in the article you mentioned. In our manuscript, we devoted an entire paragraph to chronic heart failure, including heart failure with preserved ejection fraction. There has not been such a comprehensive review before.

Additionally, we have added a summary table to the article, which should undoubtedly systematize the entire manuscript.

Reviewer 2 Report

Comments and Suggestions for Authors

This manuscript explores the multifaceted role of lactate in heart failure (HF), with a particular focus on its diagnostic and prognostic relevance in both acute and chronic HF. The topic is timely and clinically important, considering the expanding understanding of metabolic alterations in HF and the increasing interest in biomarkers beyond traditional cardiac markers. The review is well-structured, comprehensive, and provides useful insights into the emerging significance of lactate in cardiovascular pathophysiology. However, there are some areas where clarity, focus, and scientific rigor can be improved to enhance the manuscript’s impact and readability.

  1. Abstract: please clarify in the abstract that this is a narrative review rather than original research. Consider summarizing key themes or findings, such as the dual role of lactate as both a marker and a substrate, to better frame the scope of the paper.
  2. Scope and Objectives: the introduction would benefit from a clearer articulation of the manuscript's aims. At present, it reads largely as background information. Explicitly state the objective of this review—e.g., to synthesize current evidence on lactate's role in HF diagnosis, prognosis, and potential as a therapeutic target. In addition, the introduction should be shortened. It currently contains a substantial amount of general information that could be condensed or integrated into later sections.
  3. Some sections, particularly in the discussion of acute heart failure, contain repetitive statements about the association between elevated lactate and poor outcomes. These should be consolidated to streamline the narrative.
  4. Terminology and definitions: define key terms such as “hyperlactatemia,” “lactic acidosis,” and “anaerobic threshold” early and use them consistently. Also, ensure clear distinctions between AHF and CHF when discussing findings or implications.
  5. I suggest addressing the potential role of sodium-glucose co-transporter 2 (SGLT2) inhibitors in modulating lactate levels. These agents, widely used in HF management, are known to reduce oxidative stress and improve mitochondrial function, two mechanisms that could favorably influence lactate metabolism in patients with HF (cite PMID: 40245989).
  6. Conclusion: The conclusion section is comprehensive but could be shortened and more focused. Emphasize the key takeaways, such as lactate’s value in risk stratification and potential as a therapeutic target, without repeating previously stated content.
  7. Language and style: while generally well-written, some sentences are overly complex or contain minor grammatical issues. For example, "Lactate is produced mainly by glycolysis and plays a special role in the regulation of tissues and organs..." could be revised for clarity and flow. A thorough language edit is recommended to enhance readability.

Author Response

We would like to thank you for your valuable comments and the time you spent on conducting a thorough review. Based on them, we have made a change in line 68 and defined the type of literature review.

"Lactate is a biomarker that has garnered increasing attention for its potential role in the diagnosis and prognosis of heart failure. In this narrative review, we explore the physiological and pathophysiological significance of lactate in both acute and chronic heart failure. Over the past decades, numerous studies have investigated the role of lactate, particularly in critically ill patients, shedding light on its complex functions in metabolic regulation, tissue perfusion, and disease progression. This article aims to synthesize current knowledge and highlight the clinical relevance of lactate in the context of heart failure."

Additionally, we have attached a table summarizing the works we used to the article. Once again, we would like to thank you for your valuable comments.

One more we thank the Reviewer for the constructive and thoughtful comments, which significantly helped us improve the clarity and quality of our manuscript. Below we provide a point-by-point response outlining how each comment was addressed. All changes are highlighted in red in the revised manuscript.

Abstract:“Please clarify in the abstract that this is a narrative review rather than original research. Consider summarizing key themes or findings, such as the dual role of lactate as both a marker and a substrate.”
Response: We have revised the abstract to explicitly state that this is a narrative review. We also included a summary of key findings, emphasizing lactate’s dual role as both a metabolic substrate and a biomarker of disease severity. Change implemented on page 1, Abstract section (highlighted in red).
2. Scope and Objectives in the Introduction: “The introduction would benefit from a clearer articulation of the manuscript’s aims. Explicitly state the objective of this review… Also, shorten the introduction.”
Response: We shortened the introduction by removing epidemiologic and background details that were repeated later in the text. Additionally, we explicitly stated the objective of the review as: “This narrative review aims to synthesize current knowledge on the role of lactate in both acute and chronic HF, emphasizing its diagnostic, prognostic, and potential therapeutic implications.” Updated in the Introduction, page 2 (highlighted in red).
3. Repetition in the AHF Section: “Some sections, particularly in the discussion of acute heart failure, contain repetitive statements about the association between elevated lactate and poor outcomes.”
Response:
We carefully reviewed the AHF section and consolidated repetitive statements related to elevated lactate and prognosis. Multiple citations with similar outcomes were combined into a concise paragraph for clarity and efficiency.
Revisions made in Section 2 – Acute Heart Failure (highlighted in red).
4. Terminology and Definitions:
“Define key terms such as ‘hyperlactatemia’, ‘lactic acidosis’, and ‘anaerobic threshold’ early and use them consistently.”Response: Definitions of hyperlactatemia, lactic acidosis, and anaerobic threshold were added at the beginning of the AHF section to ensure clarity and consistent terminology throughout the manuscript.
Inserted early in Section 2 (highlighted in red).
5. SGLT2 Inhibitors and Lactate:“I suggest addressing the potential role of sodium-glucose co-transporter 2 (SGLT2) inhibitors in modulating lactate levels.”
Response:
We added a paragraph in the CHF section describing the emerging evidence on SGLT2 inhibitors and their potential effects on mitochondrial function and lactate metabolism, citing the suggested reference (PMID: 40245989).
Addition made in Section 3 – Chronic Heart Failure (highlighted in red).
6. Conclusion:
“The conclusion section is comprehensive but could be shortened and more focused. Emphasize the key takeaways.”Response: The conclusion was rewritten to avoid repetition and focus on the key insights: lactate’s dual role, its value in risk stratification, and its relevance as a potential therapeutic target.
Updated in Section 4 – Conclusions (highlighted in red).
7. Language and Style:
“While generally well-written, some sentences are overly complex or contain minor grammatical issues.”
Response: We performed a thorough language revision of the abstract, introduction, conclusion, and key discussion paragraphs. Sentences were shortened and simplified for clarity and better flow.
Stylistic edits implemented throughout the text, highlighted in red.

Reviewer 3 Report

Comments and Suggestions for Authors

The article provides a clear, well-documented, and timely review of the role of lactate in heart failure, addressing both pathophysiological and clinical aspects.
I suggest a few improvements to enhance the clarity and completeness of the manuscript:

- Please clarify the type of review conducted (narrative, scoping, etc.) and outline the inclusion criteria for the referenced studies.
- Consider adding a summary table comparing key studies on lactate in AHF and CHF, including patient populations, lactate thresholds, and clinical outcomes.

Author Response

Thank you very much for your valuable comments and thorough analysis of our manuscript, as well as for your time and effort. We agree that the manuscript should be expanded with a summary table, which we have created and added to the revised version. We appreciate your insightful feedback, which has helped us improve our work. Additionally, we have included information in the text on the basis of which we conducted our literature review. 
In line 68 we have changed the text to add the following information
"Lactate is a biomarker that has garnered increasing attention for its potential role in the diagnosis and prognosis of heart failure. In this narrative review, we explore the physiological and pathophysiological significance of lactate in both acute and chronic heart failure. Over the past decades, numerous studies have investigated the role of lactate, particularly in critically ill patients, shedding light on its complex functions in metabolic regulation, tissue perfusion, and disease progression. This article aims to synthesize current knowledge and highlight the clinical relevance of lactate in the context of heart failure"

Round 2

Reviewer 1 Report

Comments and Suggestions for Authors

The Authors have modified the ms according to the suggestion of this reviewer: no further comments.

Author Response

One more thank You for Your comments